# Whole-slide-imaging Cancer Metastases Detection and Localization with Limited Tumorous Data

**Yinsheng He, Xingyu Li**

**Editors:** Accepted for publication at MIDL 2023

## Abstract

Recently, various deep learning methods have shown significant successes in medical image analysis, especially in the detection of cancer metastases in hematoxylin and eosin (H&E) stained whole-slide images (WSIs). However, in order to obtain good performance, these research achievements rely on hundreds of well-annotated WSIs. In this study, we tackle the tumor localization and detection problem under the setting of few labeled whole slide images and introduce a patch-based analysis pipeline based on the latest reverse knowledge distillation architecture. To address the extremely unbalanced normal and tumorous samples in training sample collection, we applied the focal loss formula to the representation similarity metric for model optimization. Compared with prior arts, our method achieves similar performance by less than ten percent of training samples on the public Camelyon16 dataset. In addition, this is the first work that show the great potential of the knowledge distillation models in computational histopathology. The source code is publicly available at https://github.com/wollf2008/FW-RD.

**Keywords:** WSI, Tumor Detection and segmentation, Knowledge Distillation

## 1. Introduction

In the past decades, various deep-learning-based methods have been proposed to assist pathologists to detect and segment the cancer regions on histopathology slide images (Mori et al., 2013; Lee and Paeng, 2018; Morar et al., 2012; Bandi et al., 2018; Li and Ping, 2018; Tian et al., 2019). However, all of these high-accuracy cancer detection methods follow the data-driven based supervised learning paradigm, where a large number of well-annotated whole slide images (WSI) containing tumors is demanding for model generalization and robustness. To fully explore information in training data, prior arts proposes various methods to enhance tumor detection and segmentation performance. For example, Joseph et al. proposed a visual field expansion-based self-supervised method to eliminate the need for pixel-level annotations(Boyd et al., 2021). Jiaojiao et al. proposed an unsupervised cell ranking-based method that uses several pixel-level annotated cancer images supplemented by a large amount of slide-level annotated cancer images during training(Chen et al., 2019). Although these methods achieved promising performance, they still did not eliminate the great demand for cancer samples.

In this study, we tackle the problem of few-shot learning on WSIs. It should be noted that conventional few-shot learning methods on natural images cannot be directly applied to histopathological WSIs due to their huge size. In this study, we follow the conventional WSI analysis methods and extract tissue patches from WSIs for model optimization. Though we can extract thousands of image patches from few WSIs, the ratio between normal and

tumor patches is extremely unbalanced and the majority of training patches are tumor-free. To address this problem and to fully exploit knowledge provided in training samples, we incorporate the concept of anomaly detection (AD) in model design and follow the latest AD architecture, reverse knowledge distillation (RD) (Deng and Li, 2022), to tumor localization and segmentation. Such design encourages the model to learn majority patterns from normal patches. Note that the original RD model cannot be optimized using samples from different categories. To take advantages of extra yet limited tumorous patch samples in training, we further introduce a weighted similarity loss in the focal loss format to adapt the RD model to the extremely unbalanced scenario, which helps overweight the tumorous samples and enhances the sensitivity of the proposed RD-based pipeline to tumor detection and localization. We evaluate our method using the publicly accessible Camelyon16 WSI dataset and show that the proposed method only needs 1/40 of normal patches and 1/400 of tumor patches to achieve similar performance with prior arts. In addition, this is the first attempt in literature to exploit knowledge distillation in computational histopathology. Our results demonstrated the great potential of the distillation model in this field.

## 2. Related Work

**Anomaly detection** (AD) refers to identifying and localizing anomalies with limited, even no, prior knowledge of abnormality. Since animosities may vary with great diversity, it achieves the goal mainly based on learning inherent characteristics of the normal data. Among the various AD methods, generative models are the most important building block. The key idea is that generative models trained solely on normal samples can accurately reconstruct themselves but cannot do so for abnormal data (Akcay et al., 2018; Bergmann et al., 2018; Schlegl et al., 2017). However, recent studies show that many deep learning models generalize so well that even abnormal samples can be well-reconstructed(Zavrtanik et al., 2021). To address this issue, several methods, such as memory mechanism(Gong et al., 2019), image masking strategy(Yan et al., 2021), pseudo-anomaly(Pourreza et al., 2021), and knowledge distillation (Deng and Li, 2022) are introduced.

**Few-shot learning** is a type of machine learning where the goal is to learn a model that can perform well on a task with a tiny number of training examples. It is particularly useful when it is difficult or expensive to obtain a large amount of well-labeled training data. In literature, few-shot learning approaches follow two paradigms, meta-learning (Song et al., 2022; Li et al., 2021; Finn et al., 2018) and metric learning (Vinyals et al., 2016; Snell et al., 2017) In recent years, few-shot learning methods have demonstrated their utility in the medical image detection and segmentation field(Sun et al., 2022).

In this study, we utilize few-shot WSIs to train a model for tumor metastases detection and localization. Unlike the conventional few-shot learning setting where a complete set is provided, all WSIs in our study contains cancerous regions. To process the megabit information in one WSI, we follow the convention in the literature and extract image patches. Notably, tumor areas in one WSI are relatively small, which leads to an extremely unbalanced ratio of normal patches and cancerous patches. To efficiently leverage information among the obtained histopathological image patches, we adopt the concept of anomaly detection and encourage the model to learn the normal patterns shared by the few-shot WSIs.

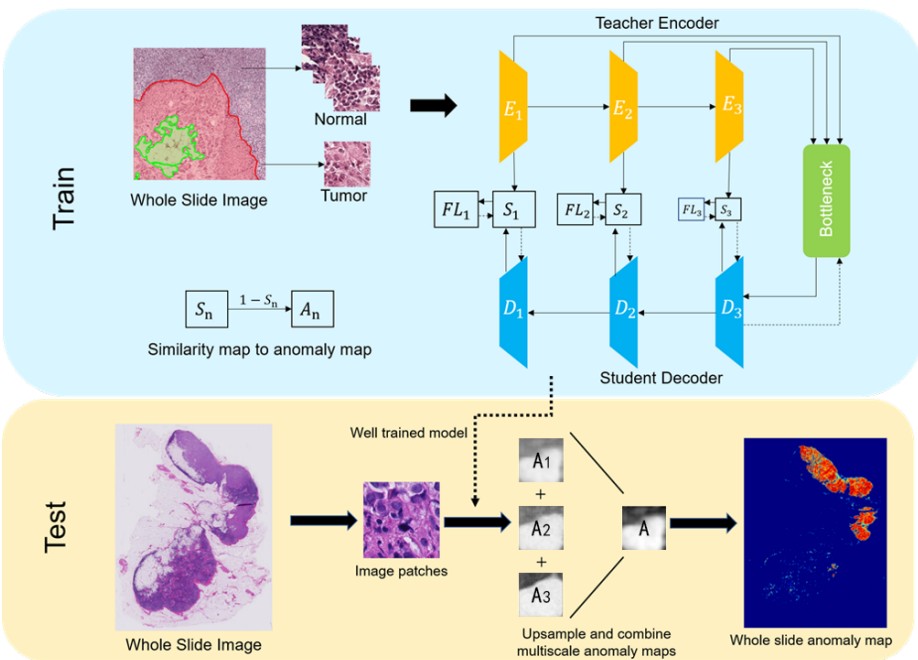

Figure 1: Framework of the proposed method, where the structure of RD model is shown in the training stage. The student net $D$ is trained to mimic the behavior of teacher net $E$ to generate similar representations in different scales for normal patches, otherwise $D$ should make the numerical feature as different as possible from $E$ for cancerous patches. During interface, we use the well-trained RD model to generate multi-scale anomaly maps $A_i$ for each patch. Then we calculate the patch-based anomaly score and combine them together for an anomaly map of the query WSI.

## 3. Methodology

We specify the proposed cancer metastases detection method using few-shot WSIs in this section. Given several pixel-level annotated WSIs that contain cancerous tissue, we follow the conventional patch-based approach to analyze WSIs. Specifically, given a WSI, we crops hundreds of small tumor patches as negative samples and thousands of small normal patches as positive samples and create a small training dataset $\mathcal{I} = \{I_1, I_2, ..., I_N\}$ with extremely unbalanced positive and negative ratio, for example, 10:1. Each image patch is associated with a 0/1 label $y_i$, where $y_i = 1$ indicates a normal patch. Our work aims to train a model on this patch set to recognize the difference between normal patches and tumor patches for tumor detection and localization in WSI.

The framework of the proposed method is depicted in Fig.1. Due to its powerful capability on normal pattern learning, we adopt the reverse distillation (RD) architecture (Deng and Li, 2022), consisting a pre-trained teacher net $E$ and a trainable student model $D$, as our backbone. Unlike the original RD method that trains the student net $D$ using anomaly-free samples only, we feed both positive and negative patches to the model in train-

ing. Note that with such an extremely unbalanced dataset, conventional data resampling strategies to address unbalanced datasets may easily fail as the whole training data is not efficiently used. To address this issue, we are inspired by the focal loss proposed in object segmentation and introduce a novel weighted similarity loss to measure the representation discrepancy in the RD model. In inference, a query WSI is also cropped into patches and the predicted anomaly maps are combined for tumor metastasis localization.

In this section, we first briefly introduce the RD model. Then we will introduce our weighted similarity loss to adapt the proposed training scenario where both positive and negative samples are available with an extremely unbalanced ratio.

### 3.1. Reverse Distillation

The RD architecture (Deng and Li, 2022) has two major components: fixed pretrained teacher encoder $E$ and trainable student decoder $D$. The trainable one-class bottleneck module is designed to transfer numerical features from teacher encoder to student decoder. The teacher encoder with WideResNet backbone (Luengo et al.) is pre-trained on imageNet and frozen, aiming to extract comprehensive representations from input patches for student net training. The student $D$ is mirror symmetric with the teacher $E$, and the purpose of it is to mimic the behavior of the teacher encoder on normal samples. Mathematically, let $E_n(I) = \{f_n \in (f_1, f_2, f_3)\}$ be the multiscale representations of an image patch $I$ in teacher net $E$, $\psi = B(E_n(I))$ denote the output of the bottleneck module, and $D(\psi) = \{f'_n \in (f'_1, f'_2, f'_3)\}$ be the multiscale features generated by student net $D$. Here, $n$ represents the $n_{th}$ block in either $E$ or $D$ and both $f_n$ and $f'_n$ have size due to their mirror symmetric architecture. In encouraging student $D$ to follow the behavior of teacher $E$ during training, a cosine similarity between $f_n$ and $f'_n$ is calculated for a similarity map $S_n(h, w)$ and the map-wise score is used as the student optimization loss.

$$S_n(h, w) = \frac{f_n(h, w)f'_n(h, w)}{||f_n(h, w)||||f'_n(h, w)||}, \tag{1}$$

where $S_n$ is in the range of [0,1]. The vector features in $f'_n$ similar to the original feature in $f_n$ will get a similarity score close to 1.

### 3.2. Weighted distillation loss

In this study, we create a training set from few-shot WSIs containing both positive and negative image patches. Though we can directly apply the original RD model to the training majority, i.e. the large amount of normal patches, this strategy wastes tumorous patches and doesn't fully utilize training data. However, since the training set $\mathcal{I}$ has extremely unbalanced positive and negative samples, we need to address two issues described as follows to adapt the RD model for our purpose.

The first question to be answered is how to accommodate both negative and positive samples in the RD model. The original RD model is trained on normal samples only. So the model training aims to minimize the representation similarity between the teacher-student pair. However, in our problem where both positive and negative samples are available, a good student model should generate similar representations to the teacher's if a patch contains normal tissue only, but generates distinct numerical features for tumorous patches.

That is, we want to minimize the similarity score $S_n$ for normal patches but maximize the loss for tumorous patches. To unify the loss function as a minimization function, the loss function for tumorous patches is modified as $1 - S_n$.

The second issue needed to be addressed is how to handle the extremely unbalance ratio between positive samples and negative samples. Since the majority of the patch training set is normal cases, the tumorous patches may be overwhelmed by normal samples and their corresponding loss may be overlooked in model training. To address this issue, we adopt the idea of focus loss in object segmentation and introduce a weighted loss to combine representation similarities of positive and negative samples as follows:

$$\mathcal{L} = -\alpha_t (1 - S_{nt}(h, w))^\gamma log(S_{nt}(h, w)), \tag{2}$$

$$\text{where} \quad S_{nt} = \begin{cases} S_n, & y_i = 1 \\ 1 - S_n, & y_i = 0. \end{cases} \quad \text{and} \quad \alpha_t = \begin{cases} \alpha, & y_i = 1 \\ 1 - \alpha, & y_i = 0. \end{cases}$$

The hyper-parameter $\alpha_t$ is used to adjust the weight between positive and negative samples so avoid overwork on one class. and the other hyper-parameter $\gamma > 1$ helps the loss function focus on tumor samples as it gets a much higher loss score than common samples. In this study, we specifically set $\alpha = 0.1$ and $\gamma = 2$.

### 3.3. Tumor Localization and detection in WSIs

With a well-trained RD model, given a patch from a query WSI, we can obtain a set of anomaly maps $A_n(h, w) = 1 - S_n(h, w)$ for $n = 1, 2, ...$, each measuring the discrepancy between the $n^{th}$-level representations in the teacher-student pair. In order to comprehensively evaluate the anomaly score of image patches in different dimensions, we up-sample the obtained anomaly maps to the same size as the input patch, then add them together to get a final anomaly map, $A(h, w) = \sum_{n=1}^{N} A_n(h, w)$. High values in the map $A(h, w)$ indicate tumor regions and the summation of the pixel-wise anomaly score in $A(h, w)$ is then compared to a pre-determined threshold for tumor detection.

## 4. Experiments

### 4.1. Experimental settings

**Dataset:** we evaluate the proposed method using the publicly Camelyon16 dataset (Bejnordi et al., 2017). Camelyon16 contains 110 tumor and 160 normal annotated WSIs for training and 81 normal and 49 tumor annotated WSIs for testing. We randomly select 10 tumor WSIs as our training data. To create the patch-level training set from the 10 WSIs, we follow the preprocess method in (Li and Ping, 2018). Specically, on $40\times$ magnification WSIs (level 0), we randomly cropped 5k $256 * 256$ patches from normal regions and 500 $256 * 256$ patches from tumor regions. For the validation purpose, we randomly cropped 2k patches from 2 WSIs. The rest 128 WSIs are used as test data. In both validation and test sets, the ratio between normal and abnormal patches is 1:1. The distribution of training and testing data is specified in Tab. 1.

**Implementation details:** We use the first three blocks of a pretrained wideResNet50 (Zagoruyko and Komodakis, 2016) architecture as the teacher encoder in our RD model. For

|  |  | Few-shot WSI setting | | |
|---|---|---|---|---|
|  |  | Train | Val. | Test |
| # of WSI | normal | 0 | 1 | 81 |
|  | tumor | 10 | 1 | 49 |
| Patches | normal | 5000 | 2000 | 10000 |
|  | tumor | 500 | 2000 | 10000 |

Table 1: Our few-shot WSI experimental setting constructed from the Camelyon16 dataset.

each block, a convolution layer with kernel size =1 and stride = 2 is used to down-sampling the data. Correspondingly, each block in the student decoder adopts a deconvolutional layer to up-sample the data with a kernel size of 2 and a stride of 2. The whole model was implemented with PyTorch and trained on NVIDIA GeForce GTX 3090. We utilize Adam optimizer(Kingma and Ba, 2014) with $\beta = (0.5, 0.999)$. The learning rate is 0.00001 and the batch size is 32. To get the best result, we set the threshold for tumor detection to 2 and train the model for 50 epochs and select the checkpoint with the best patch-level classification accuracy on the validation set.

We compare the proposed method with prior arts including HMS&MIT (Wang et al., 2016), SFCLD, TCBB (Chen et al., 2019), and the original reverse distillation model. HMS&MIT and SFCLD are the fully-supervised methods trained on the whole Camelyon16 dataset with top performance. TCBB is a few-shot learning method using 30K training patches with 1:1 normal and tumor ratio.

Following the requirements of the Camelyon16 challenge, we use the area under receiver operating characteristic(AUROC) and lesion-based free-response receiver operating characteristic(FROC) curve as our evaluation metrics.

### 4.2. Results and Discussions

Slide-level classification (AUROC) and Tumor region localization (FROC) are reported in Fig.2(a) and (b), respectively. The patch-level classification accuracy of different methods is shown in Tab.2. Our method achieves 0.9036 AUROC with only 10 pixel-level annotated WSIs. The performance of our method significantly exceeds the prior few-shot method and the baseline RD model. Fig.4.2 provides tumor localization on Test_001 for visualization. Due to some reproducible issues, we directly copied the TCBB probability map from the original paper. Though our training samples are much smaller than all prior methods, the proposed method significantly outperforms the few-shot method TBCC and the RD baseline in terms of tumor localization. Even compared to the fully-supervised methods on the complete Camelyon16 dataset, our method also achieves quite promising performance.

| Model Name | HMS&MIT | SLFCD | TCBB | RD | Our |
|---|---|---|---|---|---|
| **Patch classification accuracy** | 0.984 | 0.933 | 0.876 | 0.652 | 0.881 |

Table 2: Patch level classification accuracy on Camelyon16

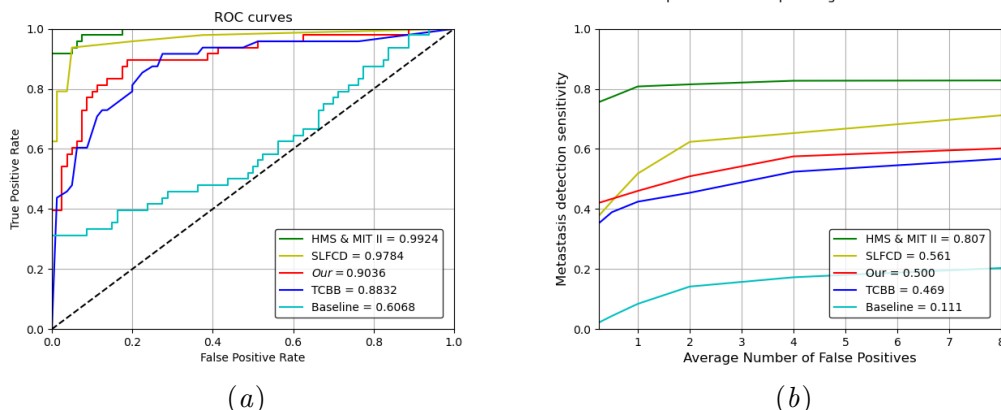

(a)                                           (b)

Figure 2: (a) ROC curves and (b) FROC curves on Camelyon16. HMS&MIT is the winner of the Camelyno16 challenge and SLFCD is the latest study. Both of them used the entire training set. TCBB is the latest few-shot method on Camelyon16, utilizing 3 times more data than our method in training.

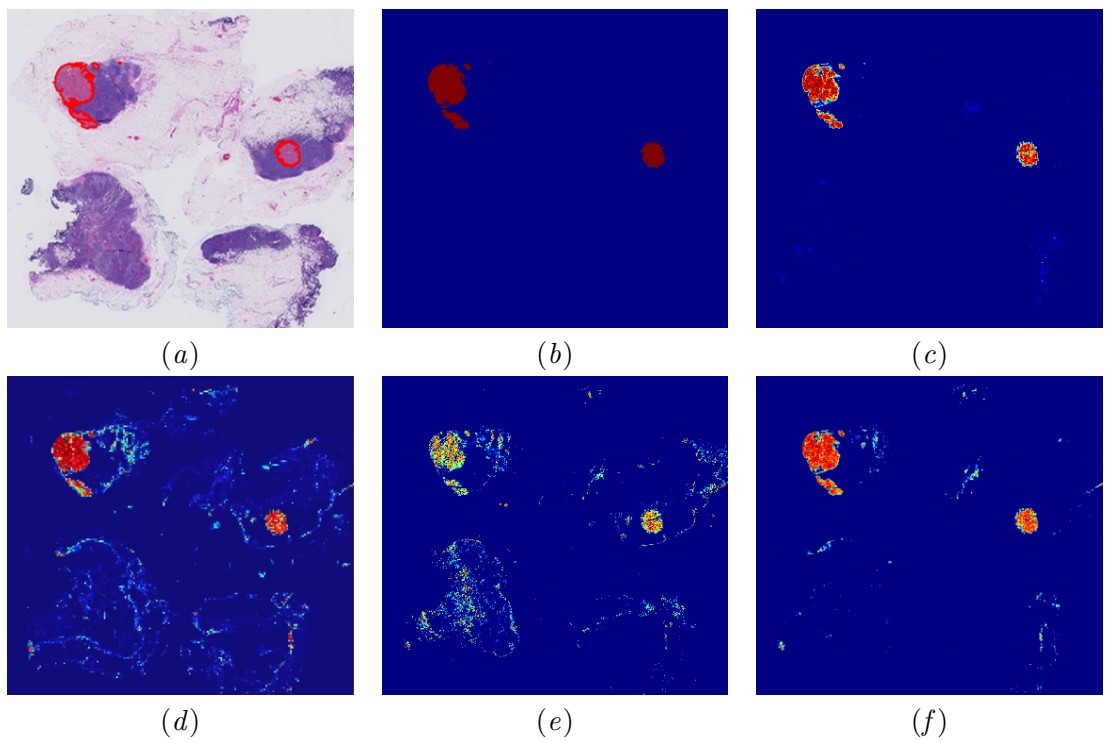

Figure 3: (a) Original WSI Test_001 in Camelyon16, (b) the ground truth of tumor area, and tumor localization by (c) SFCLD, d) TCBB, (e) the original RD, and (f) our method.

| Number of abnormal patches | 0 | 5 | 10 | 50 | 100 | 500 | 1000 |
|---|---|---|---|---|---|---|---|
| Patch classification Accuracy | 0.646 | 0.759 | 0.784 | 0.812 | 0.843 | 0.881 | 0.879 |

Table 3: Ablation on the number of tumor patches in model optimization.

### 4.3. Ablation on Weighted Loss

The weighted distillation loss mitigates the degradation of classification performance caused by data imbalance. In this study, we compare the weighted and unweighted loss. Under the same setting, the patch classification accuracy with unweighted loss reduces to 83.8%, which is a 4.3% performance drop compared to the weighted loss.

### 4.4. Ablation on Training Samples

This ablation studies how the amount of training data affects the performance of our method. To this end, we first vary the number of abnormal patches in training and monitor the patch-level classification accuracy following the previous study in (Wang et al., 2016). To ensure the fairness, all experiments use the same normal patch set, and all small negative patches are subsets of the original tumor training samples. Under each set, we simply modify $\alpha$ so that it follows the ratio between normal and abnormal patches and report the patch-level classification accuracy in Table3. When the number of abnormal patches is 0 and only 5K normal patches are available for training, our method is identical to the original RD model. Few more abnormal patches (such as 5 or 10) could noticeably improve the model's performance.

Second, we evaluate the model with a balanced dataset by downsampling the normal patches. To this end, we randomly resample 500 normal patches (same as the number of tumor patches in our main experiment) and the classification accuracy reduces to 86.9%. This performance drop is due to the information loss in normal pattern learning.

For a comprehensive investigation of the proposed method, we also quantitatively evaluate our model under a fully-supervised setting and compare its performance to the prior supervised methods. Specifically, we crop 200,000 patches from the entire 270 WSIs in the Camelyon16 training set, half being normal patches and the other half being tumor patches. We set the hyperparameters $\alpha = 0.5$ and $\gamma = 2$. The classification accuracy under this setting reaches 93.0%, which is still lower than those state-of-art methods but very close (e.g. 93.3% FOR SLFCD).

## 5. Conclusions

In this work, we proposed a reverse distillation-based WSI few-shot learning method to localize tumor regions in WSIs. To address the unbalanced issue in training set, our method incorporated the concept of anomaly detection in model design and encouraged the model to learn normal patterns from the majority of training set. To further exploiting information in abnormal patches, we introduced a focus loss similar function to upweight the minority samples in model optimization. The results indicated that our model could identify the tumorous regions with promising performance and achieved test AUC scores greater than 0.9.

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
