# OpenReview forum: "Whole-slide-imaging Cancer Metastases Detection and Localization with Limited Tumorous Data"
_MIDL.io/2023/Conference — MIDL 2023 Poster_

### Official Review · Reviewer_FaXP · 2023-02-01

**Confidence:** 4
**Preliminary Rating:** 3
**Recommendation:** Poster

**Summary:**

The paper proposes a combination of anomaly detection and few-shot learning for detection of metastases in histopathology images of lymph nodes of breast cancer patients. The base method is the recently proposed knowledge distillation anomaly detection approach, which is modified to be able to make use of limited number of annotated positive examples (anomalies).

Overall, anomaly detection methods are, in my view, underexplored for analysis of histopathology images. As such, I find this paper to be very interesting for the computational pathology community.

**Strengths:**

Relatively straightforward methodology based on recent advances from the computer vision field. The proposed modification of the reverse destination approach is well motivated. It is evaluated on a publicly available dataset.

**Weaknesses:**

Methodological novelty is rather limited, so I see this more as an application paper (this is not a weakness, but it should be noted). The major weakness that I see lies in the experimental setup, more specifically in the way that the positive examples (anomalies) were sampled. While the total number of included positive patches is very low, it seems that they were samples from all available tutor slides. This goes against the the use case of few-shot learning - while only a limited number of patches are samples, they are sampled from all available slides. This reduced the practical applicability of the method: it is not applicable (or at least we do not know how well it performs) for the case when many negative slides are available, but there are only a few (2-5) slides with tumor.

**Deanonymize Review:**

no

**Detailed Comments:**

- There are many small typos in the text that should be corrected.
- The references are not in the correct format (journal/venue is missing).

**Paper Type:**

validation/application paper

**Questions To Address In The Rebuttal:**

I would really like to see a true "few shot" experimental setup where a limited number of patches is sampled from a limited number of slides. As there can be a lot of tissue appearance variability, it can be expected that limited the number of available slides will have significant effects on the performance.

---

### Official Review · Reviewer_Vs4H · 2023-02-01

**Confidence:** 4
**Preliminary Rating:** 4
**Recommendation:** Poster

**Summary:**

Training fully supervised localization and detection model for WSIs relies on more training samples and well-annotated labels.  To release the model so that it can work on the few-shot setting for WSI, this paper propose a patch-based analysis baseline based on the reverse knowledge distillation architecture. Furthermore, to account for the issue of extremely imbalance in normal and tumorous samples, this paper proposed a weighted loss inspired from the focal loss. To this end, this paper outperforms the previous few-shot methods with less training samples.

**Strengths:**

Learning under impect supervision is an important field for the WSIs, where both the annotation and training samples can be limited. This paper tackles the important issue via a few shot learning setting. By introducing the methods from the anomaly detection to the few-shot setting of WSI, this paper further adapts the methods so that it can tackle the highly imbalanced class scenario. To this end, sufficient experiments and ablation study show the superiority of this paper.

**Weaknesses:**

1. in the part of the experiment, the author mentioned that the baseline model is the RD model. It is not totally clear to me what exactly the author means. Is it the model that distilled without the abnormal patches (just like that in section 4.3), or it is the original wideResnet pretrained on the ImageNet without distilling? More details are needed.


2. The author mentioned that the weighted loss is important for the class imbalance setting. This is intuitive, however, an ablation study can be provided to validate such a claim in a quantitative way.

3. Though the paper aims to work in the few-shot setting, it would be interesting to see how the methods scale with more training data as we can tell there is still a gap between this method and the fully supervised one. For example, if using the same amount of data as the fully-supervised methods, how does the proposed method perform? Regardless of the outcome, the significance of this paper won't be diminished. Including such an experiment would only strengthen this paper.

**Deanonymize Review:**

no

**Detailed Comments:**

See the comment above

**Paper Type:**

both

**Questions To Address In The Rebuttal:**

See the weakness section. I would consider raising the score if the author can resolve the following concerns.


please

1: provide more details on the implementation of the baseline methods.

2: provide the ablation study on the weighted loss part.

---

### Official Review · Reviewer_Tdgm · 2023-02-05

**Confidence:** 4
**Preliminary Rating:** 4
**Recommendation:** Poster

**Summary:**

In this paper, the authors propose to apply a modified version of a reverse knowledge distillation (RD) framework to detect and localize tumors in whole-slide histopathological imagery. Rather than use RD directly, which is trained on only normal samples (with abnormal samples having maximum dissimilarity), the authors propose to train on both normal and abnormal samples, and adopt a focal loss to deal with the class imbalance.

**Strengths:**

1. The proposed method is one of the first, if not the first, to apply knowledge distillation to histopathology imagery.
2. The authors demonstrate that significant advantage can be gained from utilizing both positive and negative samples with the adapted focal loss (RD: 0.652 vs proposed: 0.881).
3. Overall the paper is well written, easy to follow, and well motivated.
4. The ablation over samples is helpful.

**Weaknesses:**

The normal focal loss was first introduced for object detection. In that use, there are both positive and negative samples. The authors suggest they are modifying focal loss to do a novel cosine similarity version. It is unclear to the reviewer, how exactly is the proposed loss different than the original focal loss? If there is a novel modification, there should be an ablation study directly showing the contribution of this modification from the original focal loss.

**Deanonymize Review:**

no

**Paper Type:**

validation/application paper

**Questions To Address In The Rebuttal:**

1. Are the authors just applying focal loss, or are they making some novel change to the focal loss formulation? If so, a ablation compared against the original focal loss would be extremely valuable. If not, the paper should make it clear that focal loss is being applied and not "a novel weighted loss function." I would be willing to raise the review to accept if this is addressed (either by including the ablation or reworking the language to say apply and not claim a novelty that doesn't exist).

---

### Official Review · Reviewer_tV1z · 2023-02-07

**Confidence:** 4
**Preliminary Rating:** 4
**Recommendation:** Poster

**Summary:**

The authors proposed a patch-based analysis pipeline for tumor localization and detection problem under the setting of few labeled whole-slide images. The proposed approach is built upon a reverse knowledge distillation framework and introduces a focal loss-like loss function to address the class imbalance issue. The results show that the proposed approach can achieve similar performance compared to prior arts with less training samples.

**Strengths:**

- The paper is well-written and the motivation is very clear.
- This paper shows that for the first time, it is feasible to use a reverse distillation method for histopathology under the few label setting.
- The proposed modification on the vanilla reverse distillation is interesting and seems effective for anomaly detection tasks with imbalanced positive and negative samples.


**Weaknesses:**

- There is a lack of comparison to other anomaly detection techniques in the experiments.
- It is unclear how effective the proposed weighted distillation loss is. An ablation study that compares the weighted and unweighted loss should be conducted to further validate the effectiveness.

**Deanonymize Review:**

no

**Detailed Comments:**

- Please correct the typos in the manuscript. E.g., ' achived he ' in the conclusion section.
- It would be interesting and more clear to show the visualization  A1, A2 and A3 to help readers to better understand each anomaly map.

**Paper Type:**

methodological development

**Questions To Address In The Rebuttal:**

- Please justify the effectiveness of the weighted distillation loss. It remains unclear that (1) how the weighted distillation loss is better than the unweighted counterparts and (2) why positive samples and negative samples are initially taken with a ratio of 10:1. How would directly downsample the number of positive samples to reach a balance between positive and negative samples affect the performance?

---

### Meta-Review · Area_Chair_9pAh · 2023-02-23

**Recommendation:** Accept (Poster)
**Confidence:** 4

**Metareview:**

This paper proposed a patch-based analysis pipeline for tumor localization and detection problem from few labeled whole-slide images.

Pros:
1. Paper is well-written and motivation is clear.
2. The proposed method is one of the first, if not the first, to apply knowledge distillation to histopathology imagery.
3. sufficient experiments to show the superiority of this paper.

Cons:
Lack of some comparison with other techniques such as anomaly detection.
More detailed ablation studies.
Clear elaboration on the difference of loss functions.
More details on the implementation of the baseline methods.
Typos in the paper, etc.

All reviewers firmed the merits of the paper and the concerns can be addressed properly in the final version. Therefore, a decision of accept is recommended.